

# Seed germination and early seedling survival of the invasive species *Prosopis juliflora* (Fabaceae) depend on habitat and seed dispersal mode in the Caatinga dry forest

Clóvis Eduardo de Souza Nascimento[1,2],
Carlos Alberto Domingues da Silva[3,4], Inara Roberta Leal[5],
Wagner de Souza Tavares[6], José Eduardo Serrão[7], José Cola Zanuncio[8]
and Marcelo Tabarelli[5]

[1] Centro de Pesquisa Agropecuária do Trópico Semi-Árido, Empresa Brasileira de Pesquisa Agropecuária, Petrolina, Pernambuco, Brasil
[2] Departamento de Ciências Humanas, Universidade do Estado da Bahia, Juazeiro, Bahia, Brasil
[3] Centro Nacional de Pesquisa de Algodão, Empresa Brasileira de Pesquisa Agropecuária, Campina Grande, Paraíba, Brasil
[4] Programa de Pós-Graduação em Ciências Agrárias, Universidade Estadual da Paraíba, Campina Grande, Paraíba, Brasil
[5] Departamento de Botânica, Universidade Federal de Pernambuco, Recife, Pernambuco, Brasil
[6] Asia Pacific Resources International Holdings Ltd. (APRIL), PT. Riau Andalan Pulp and Paper (RAPP), Pangkalan Kerinci, Riau, Indonesia
[7] Departamento de Biologia Geral, Universidade Federal de Viçosa, Viçosa, Minas Gerais, Brasil
[8] Departamento de Entomologia/BIOAGRO, Universidade Federal de Viçosa, Viçosa, Minas Gerais, Brasil

Corresponding author
Marcelo Tabarelli, mtrelli@ufpe.br

## ABSTRACT

**Background:** Biological invasion is one of the main threats to tropical biodiversity and ecosystem functioning. *Prosopis juliflora* (Sw) DC. (Fabales: Fabaceae: Caesalpinioideae) was introduced in the Caatinga dry forest of Northeast Brazil at early 1940s and successfully spread across the region. As other invasive species, it may benefit from the soils and seed dispersal by livestock. Here we examine how seed dispersal ecology and soil conditions collectively affect seed germination, early seedling performance and consequently the *P. juliflora* invasive potential.

**Methods:** Seed germination, early seedling survival, life expectancy and soil attributes were examined in 10 plots located across three habitats (flooding plain, alluvial terrace and plateau) into a human-modified landscape of the Caatinga dry forest (a total of 12,000 seeds). Seeds were exposed to four seed dispersal methods: deposition on the soil surface, burial in the soil, passed through cattle (*Boss taurus*) digestive tracts and mixed with cattle manure and passed through mule (*Equus africanus asinus × Equus ferus caballus*) digestive tracts and mixed with mule manure. Seeds and seedlings were monitored through a year and their performance examined with expectancy tables.

**Results:** Soils differed among habitats, particularly its nutrient availability, texture and water with finely-textured and more fertile soils in the flooding plain. Total seed germination was relatively low (14.5%), with the highest score among seeds buried in the flooding plain (47.4 ± 25.3%). Seed dispersal by cattle and mule also positively

impacted seed germination. Early seedling survival rate of *P. juliflora* was dramatically reduced with few seedlings still alive elapsed a year. Survival rate was highest in the first 30 days and declined between 30 and 60 days with stabilization at 70 days after germination in all seed treatments and habitats. However, survival and life expectancy were higher in the flooding plain at 75 days and lower in the plateau. *Prosopis juliflora* seedling survival and life expectancy were higher in the case seeds were mixed with cattle manure.

**Synthesis:** *Prosopis juliflora* seeds and seedlings are sensitive to water stress and habitat desiccation. Therefore, they benefit from the humid soils often present across human-disturbed flooding plains. This plant also benefits from seed deposition/dispersal by livestock in these landscapes, since cattle manure represents a nutrient-rich and humid substrate for both seeds and seedlings. The quality of the seed dispersal service varies among livestock species, but this key mutualism between exotic species is due to the arillate, hard-coated and palatable seeds. *Prosopis juliflora* traits allow this species to take multiple benefits from human presence and thus operating as a human commensal.

## INTRODUCTION

Biological invasion is an important threat to tropical biodiversity and ecosystem functioning. Increased disturbance by ever-growing human populations will make terrestrial tropical biotas more vulnerable to invasions (*Simberloff et al., 2013*; *Roy et al., 2014*; *McGeoch et al., 2016*). This increased vulnerability is due to alien species benefiting from human disturbances, such as land cleaning and soil degradation by agriculture and livestock. Humans and their commensals, whether intentionally or not, favor alien species by creating altered or novel habitats (*Almeida et al., 2015*; *Jauni, Gripenberg & Ramula, 2015*; *Bellard, Cassey & Blackburn, 2016*; *Malavasi et al., 2016*).

The mechanisms driving successful invasion by alien species and those restricting them to particular habitats or conditions are key topics in invasion science (*Blackburn et al., 2014*; *Catford & Jansson, 2014*; *Hulme, 2015*). The life-history strategy or trait package exhibited by alien species affect their competitive performance or adaptability, while environmental conditions, disturbance regimes, habitat degradation and the structure of native communities (e.g., patterns of species richness and functional composition) are external forces controlling invasion success (*Gilioli et al., 2014*; *Goia, Ciocanea & Gavrilidis, 2014*; *Banerjee & Dewanji, 2017*). The intrinsic and external factors collectively define the potential for successful invasion and delimit its ecological context, the geographic coverage and potential damage to native biodiversity. Successful invasions only occur when alien species can overcome external forces (*Dalmazzone & Giaccaria, 2014*; *Li et al., 2014*; *Svenning et al., 2014*). These factors explain why a small fraction of

introduced alien species become invasive regardless of ecosystem type or habitat integrity (*Kalusová et al., 2013*; *Van Wilgen & Richardson, 2014*; *Novoa et al., 2015*).

Successful invasion of tropical plant species usually relies on the key life-history traits across all life-cycle stages (*Chapple, Simmonds & Wong, 2012*; *Malíková, Mudrák & Klimešová, 2012*; *Mullah et al., 2014*), such as vegetative reproduction, massive seed production, effective seed dispersal, high germination success in a wide range of environmental conditions, fast growth and high phenotypic plasticity (*Moravcová et al., 2015*; *Van Kleunen, Dawson & Maurel, 2015*; *Moran, Reid & Levine, 2017*). The effective colonization of human-degraded habitats usually requires dealing with physical stress, particularly reduced soil nutrients and water availability (*Boudiaf et al., 2013*; *Pérez et al., 2015*; *Rathore et al., 2015*). The naturalization and long-term establishment of invasive species depend on integrating adaptive traits as functional strategies (*Guo et al., 2018*). However, the relative contribution of each trait or strategy vary from case to case (*Hulme & Barrett, 2013*; *Perkins & Nowak, 2013*; *Rai, 2015*).

The Caatinga of Northeast Brazil is one of the largest blocks (nearly one million km$^2$) and the world most species-rich seasonally dry tropical forest (*Da Silva, Leal & Tabarelli, 2017*). The regional vascular flora reaches 3,000 species with one third endemic, including a myriad of Cactaceae species—making this region the second diversity cactus species center globally (*Terra et al., 2018*; *Silva & Souza, 2018*; *Apgaua et al., 2018*). The Caatinga was inhabited by hunter-gather people for thousands of years (*Leal et al., 2005*; *Mamede & De Araújo, 2008*; *Da Silva, Leal & Tabarelli, 2017*). The Europeans arrived in the 16th century and the Caatinga dry forest has been converted into human-modified landscapes by a combination of extensive cattle ranching, small-scale subsistence farming and exploitation of forest products such as firewood, fodder, timber, wood for charcoal, fruits and bushmeat (*Silva et al., 2014*; *Leal, Andersen & Leal, 2014*; *Ribeiro et al., 2015*; *Da Silva, Leal & Tabarelli, 2017*). Nearly 10 million m$^3$ of firewood and charcoal are obtained per year from native species, while goat (i.e., the exotic *Capra aegagrus* subspecies *hircus* L., 1758; Artiodactyla: Bovidae) herds feeding on native vegetation exceed 16 million heads (*Ribeiro et al., 2015*; *Rito et al., 2017*; *Sfair et al., 2018*). The Caatinga supports around 28 million people, making it one of the most populated semiarid regions and one of the most degraded/vulnerable Seasonally Dry Tropical Forests (SDTFs) globally (*Oliveira et al., 2016*; *Moro et al., 2016*; *Rito et al., 2017*; *Schulz et al., 2017*; *Da Silva, Leal & Tabarelli, 2017*).

The alien flora inhabiting the Caatinga includes 205 species of 135 genera and 48 families, including 61 Poaceae and 33 Fabaceae invasive species (*Almeida et al., 2015*). This diverse alien flora includes the evergreen tree *Prosopis juliflora* (Sw) DC (Fabales: Fabaceae: Caesalpinioideae), one of the world top 100 undesired species (*Nascimento et al., 2014*; *Almeida et al., 2015*). This species is native to the Caribbean, Central America and South America, but has been intentionally introduced worldwide for its economic value, utility to rural populations and ecological rusticity due to its fast growth, ability to fix nitrogen and tolerance to arid conditions and saline soils (*Nascimento et al., 2014*; *Ilukor et al., 2016*; *Walkie et al., 2016*).

*Prosopis juliflora* was introduced in the Caatinga at early 1940s as a food source for livestock and successfully spread across the region by invading and forming monospecific

stands, particularly across former agricultural lands along river banks and flooding plains (*Nascimento et al., 2014*; *Almeida et al., 2015*; *Santos & Diodato, 2015*; *Oliveira et al., 2017*). *Prosopis juliflora* forms dense stands, apparently excluding or impelling the reestablishment of species-rich assemblages of tree and shrub species, particularly on river banks degraded by shifting cultivation and livestock overgrazing (*Pegado et al., 2006*; *Oliveira et al., 2012*; *Nascimento et al., 2014*). Caatinga dry forests along river banks are structurally complex due the presence of large tree species (e.g., *Libidibia ferrea* (Mart. ex Tul.) L.P. Queiroz, *Piptadenia stipulacea* (Benth.) Ducke (Fabales: Fabaceae) and *Tabebuia aurea* (Silva Manso) Benth. & Hook.f. ex S. Moore (Lamiales: Bignoniaceae)) and support a diverse flora, including endemic species such as the cactus *Cereus jamacaru* DC. and *Pilosocereus gounellei* (F.A.C. Weber ex K. Schum.) Luetzelb. (Caryophyllales: Cactaceae). *Prosopis juliflora*, as other invasive species, may benefit from the soils and seed dispersal by livestock (*Kebede & Coppock, 2015*; *Pasha et al., 2015*; *Alvarez et al., 2017*) but the ultimate forces permitting successful invasion of Caatinga dry forest by this plant needs further studies (*Shackleton, Le Maitre & Richardson, 2015*; *Abdulahi, Ute & Regasa, 2017*; *Naudiyal, Joachim & Stefanie, 2017*).

In this study we investigate seed germination, early seedling survival and life expectancy in *P. juliflora*, as well as soil attributes across the three main habitats covered by Caatinga dry forest to understand how seed dispersal ecology and soil conditions collectively affect seedling performance and consequently the invasive potential of *P. juliflora*. Seeds exposed to four seed dispersal treatments were deposited across flooding plain, alluvial terrace and plateau habitats, with seed and seedling fate being monitored for a year. The potential mechanisms behind differential seedling performance, especially a positive synergism between human disturbance, key ecological services provided by livestock and the life-history traits exhibited by *P. juliflora* was studied.

## MATERIALS AND METHODS

### Study site

The study was carried out in 10 sites across a 3.5-km$^2$ human-modified Caatinga landscape (9°00′ S, 40°13′ W; 377 m altitude) covering three habitats: flooding plain, alluvial terrace, and plateau (Figs. S1 and S2). This landscape is typical of the Caatinga, with farming households devoted to livestock production with animals raised extensively and feeding on the Caatinga vegetation plus small patches devoted to subsistence agriculture and remnant patches of Caatinga dry forest; that is, the traditional Caatinga land use (*Sampaio & Costa, 2011*). The landscape stretches over sedimentary basins, mountains, plateaus and ravines covered by cambisols, eutrophic podzols, lithosols, non-calcic brown soils and planosols along the São Francisco River valley (*Razanamandranto et al., 2004*). The river terrace has alluvial deposits from the valley slopes with sedimentary clay, sandy or silty material in stratified silt layers (*Miranda et al., 2014*). The flooding plain, with slopes between 0° and 2° (*Miranda et al., 2011*), consists of recent sediments from terraces (*Babawi, Campbell & Mayer, 2016*). The alluvial terrace, also called the slope, consists of flat areas or benches, usually situated above the river level, with gravel or thick sediment forming ancient terraces (*Mukherjee, Velankar & Kumara, 2017*). The plateau is a flat

terrain covered by a sedimentary clay mantle, which spreads following the river terraces (*Silva et al., 2008*; *Ferraz, Rodal & Sampaio, 2003*). The regional climate is hot semi-arid, with mean annual temperature of 26.3 °C and relative humidity of 61.7%. The 570 mm annual rainfall is concentrated between January and April (*Ramos et al., 2011*). The focal landscape was completely covered by Caatinga dry forest prior to European settlement (*Silva et al., 2008*). Regionally, Cactaceae, Euphorbiaceae and Fabaceae are the most species-rich plant families (*Leal et al., 2005*; *Silva et al., 2008*).

## Soil attributes

Soil attributes were estimated based on 10 soil samples per habitat type (flooding plain, alluvial terrace and plateau). Soil moisture was determined twice a week at 20 cm depth from March 2014 to February 2015 (*Francesca et al., 2010*). The soil samples were weighed to determine the wet mass (mw, g), dried in an oven at 105 °C for 24 h and weighed again to determine their dry mass (md, g). Soil moisture was determined by $H(g, g^{-1}) = (md - mh) \div md$ or $H(\%) = (md - mh) \div md \times 100$. Physical (field capacity, sand, silt and clay content) and chemical soil attributes (soil organic matter, phosphorus, potassium, calcium, magnesium, sodium, aluminum and cation exchange capacity (CEC)) were also obtained via 10 soil samples per habitat type, with samples at 0–20 cm depth (*Lopes et al., 2013*). Soil analyses were performed in the Laboratory for Soil, Water and Plant Tissue Analysis at EMBRAPA Semi-Arid according to standard international protocols (*Marques et al., 2007*; *Galindo et al., 2008*).

## Seed germination and seedling performance across habitats

*Prosopis juliflora* seeds were exposed to four seed dispersal treatments as follows: seeds with artificially broken dormancy deposited on the soil surface (T1); seeds with artificially broken dormancy buried in the soil to a depth of 1.0 cm (T2); seeds that passed through the digestive tract of cattle (*Bos taurus* subspecies *indicus* L., 1758; Artiodactyla: Bovidae) and mixed with cattle manure (hereafter cattle-dispersed seeds) (T3) and seeds that passed through the digestive tract of mules (donkey, *Equus africanus* subspecies *asinus* L., 1758 × mare, *Equus ferus* subspecies *caballus* L., 1758; Perissodactyla: Equidae) and mixed with mule manure (hereafter mule-dispersed seeds) (T4). These four seed dispersal treatments resemble natural seed dispersal modes and fates in the Caatinga, as ripe fruits can be (1) consumed by livestock and deposited on the soil surface immersed in manure; (2) remain intact on the ground and as the pod rots, seeds are deposited on the soil surface and (3) carried by runoff and deposited in soil sediments (*Dos Santos et al., 2016*). These fates need further studies, although ripe pods are consumed by livestock including cattle, goats, horses and mules (*Nascimento et al., 2014*). Seeds were manually collected from mature fruits from several trees collected on the ground throughout 2013. Seeds were used to feed livestock first and their manure collected.

   *Prosopis juliflora* seeds were randomly sown on 120 1.4 m$^2$-field plots across the three habitats (40 plots per habitat, Fig. S1). One thousand seeds were sown per treatment and habitat, totaling 12,000 seeds (four treatments × three habitats × 1,000 seeds per treatment). Seed dispersal treatments were set up in mid-March (i.e., rainy season) of 2014.

Seeds were collected from several trees and animal dungs in the year preceding sowing experiments. Seed dormancy was broken artificially by making a small incision with blunt scissors on the opposite side of the micropylar region, while those that had passed through the animal digestive tract were extracted from dried dung (*Nascimento et al., 2014*). Seeds obtained from trees were sown at 10 × 10 cm spacing and those collected from the manure were separated, counted, mixed with 500 mL of the respective animal dung and sown at 10 × 10 cm spacing. Seed germination and seedling survival were evaluated for each of the three habitats every 5 days during the first 30 days and every 15 days thereafter for 1 year. *Prosopis juliflora* germination is epigeal (*Dube, Mlambo & Sebata, 2010*) and sprouted seeds (with cotyledons) were marked with toothpicks and protected with wire netting to prevent animal damage. Seeds with their first cotyledon open were classified as seedlings.

## Life expectancy and survival tables

Life expectancy and survival tables of *P. juliflora* were calculated (*Bogino & Jobbágy, 2011*) for each dispersal treatment and habitat considering age classes of 15 days ($x = 15$). The number of *P. juliflora* survivors at the beginning of each age class ($L_x$) and at the germination and seedling life stages were estimated based on survival ($lx$) in these stages (*Caswell, 1996*) as follow: (a) number of dead individuals during the age class $x$ ($dx$): $dx = L_x - L_{x+1}$; (b) mortality ratio for the age class $x$ ($qx$): $qx = d_x \div L_x$; (c) survival rate during age $x$ ($sx$): $sx = 1 - q_x$; (d) age structure ($Ex$) as the number of alive plants from each age class: $Ex = (L_x + L_{x+2}) \div 2$; (e) cumulative number of living individuals ($T_x$) by: $Tx = \sum_{j \geq x}^{y} Ex$ where $j$ represents any age greater than or equal to that of class $x$; and (f) life expectancy for individuals of age class $x$ ($ex$): $ex = T_x \div L_x$.

## Data analysis

Differences in soil attributes were examined via analysis of variance and Tukey's HSD (honestly significant difference) test ($p < 0.05$) (*Tukey, 1949*). Non-normal (non-parametric) data obtained from the three habitats were compared using the Kruskal–Wallis test ($p < 0.05$) (*Wallis, 1952*). Differences on average seed germination (%) across habitats and seed dispersal modes were examined via a two-way ANOVA followed by Tukey's HSD test. Number of germinated seeds was square-rooted transformed prior analysis to approach data normality. As habitat and soil moisture were autocorrelated, soil moisture was not adopted as an explanatory variable for seed germination and seedling survival/life expectancy. All tests were run in BioEstat 5.0 program (*Ayres et al., 2007*).

# RESULTS

## Soil physical attributes

Caatinga habitats differed in terms of soil physical attributes. Field capacity was higher in the flooding plain ($F = 8.5454$, df = 2, $p = 0.0016$; Fig. 1A), with this habitat and the terrace supporting higher soil moisture ($H = 12.5507$, df = 2, $p = 0.0019$) than the plateau (Fig. 1B). Soil clay content was higher in the flooding plain followed by the plateau and the

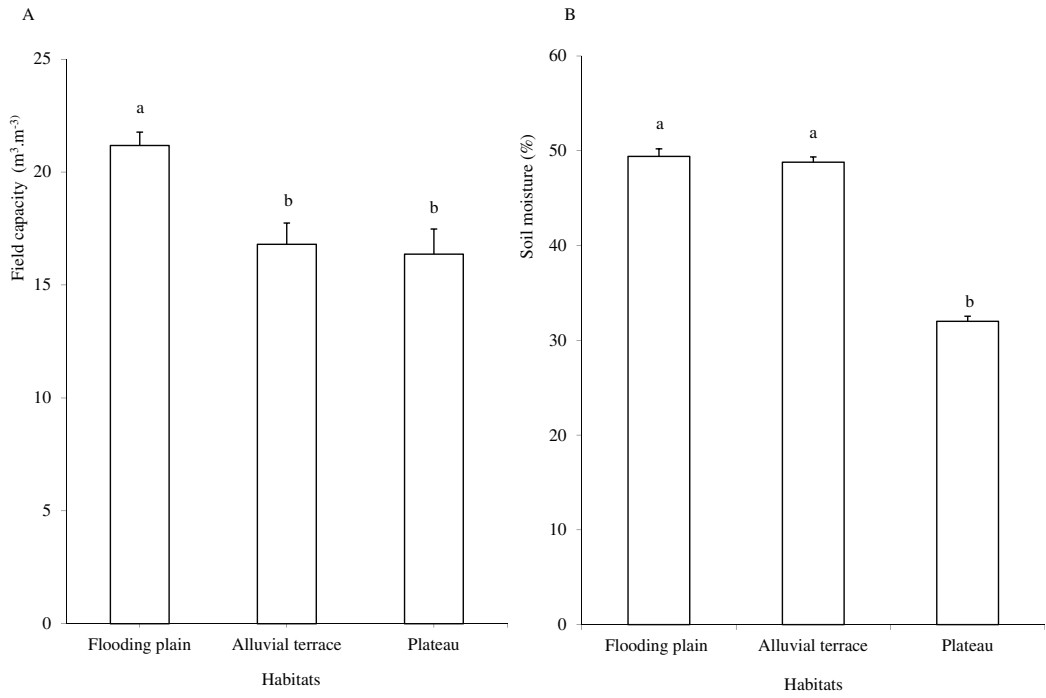

**Figure 1 Field capacity (A) and soil moisture (B) in flooding plain, alluvial terrace and plateau habitats, at 0–20 cm depth in a Caatinga dry forest, Brazil.** Tukey's Honest Significant (field capacity) and Kruskal–Wallis (soil moisture) tests. Columns followed by the same letter are similar ($p < 0.05$). Error bars indicate standard deviation.

alluvial terrace ($H = 9.0142$, df = 2, $p = 0.011$). The silt score was also highest in the flooding plain as compared to plateau ($F = 6.6178$, df = 2, $p = 0.0048$). However, soil did not differ in terms of sand content ($F = 8.2452$, df = 2, $p = 0.0019$; Fig. 2A).

## Soil chemical attributes

Overall soils varied little within each habitat, but they greatly differed in terms of chemical attributes associated with soil fertility across the habitats. We shall mention a higher content of phosphorus ($H = 16.935$, df = 2, $p = 0.0002$), potassium ($H = 16.87$, df = 2, $p = 0.0002$), and CEC ($H = 12.18$, df = 2, $p = 0.0023$) in the flooding plain as compared to plateau (Fig. 2B), while Al content was higher in the plateau ($H = 21.618$, df = 2, $p = 0.0001$). Overall, terrace soils exhibited intermediated scores, with the exception of organic matters as these soils presented reduced contents ($H = 17.8655$, df = 2, $p = 0.0001$).

## Seed germination

From a total 12,000 seeds, only 14.5% (1,746 seeds) germinated and seed germination varied according to both habitat and seed dispersal mode (Tables 1 and 2); that is, a significant interaction between habitat and seed dispersal mode. Considering habitats, higher average germination occurred among those seeds buried in the flooding plain (47.4 ± 25.3%; mean ± SD), and terrace (33.4 ± 25.8%), while seeds in the plateau experienced the lowest rates, from 0.7% to 12.9%. Among seed dispersal modes, seeds in the soil surface exhibited the lowest scores (0.7–5.3%), in contrast to the positive impacts

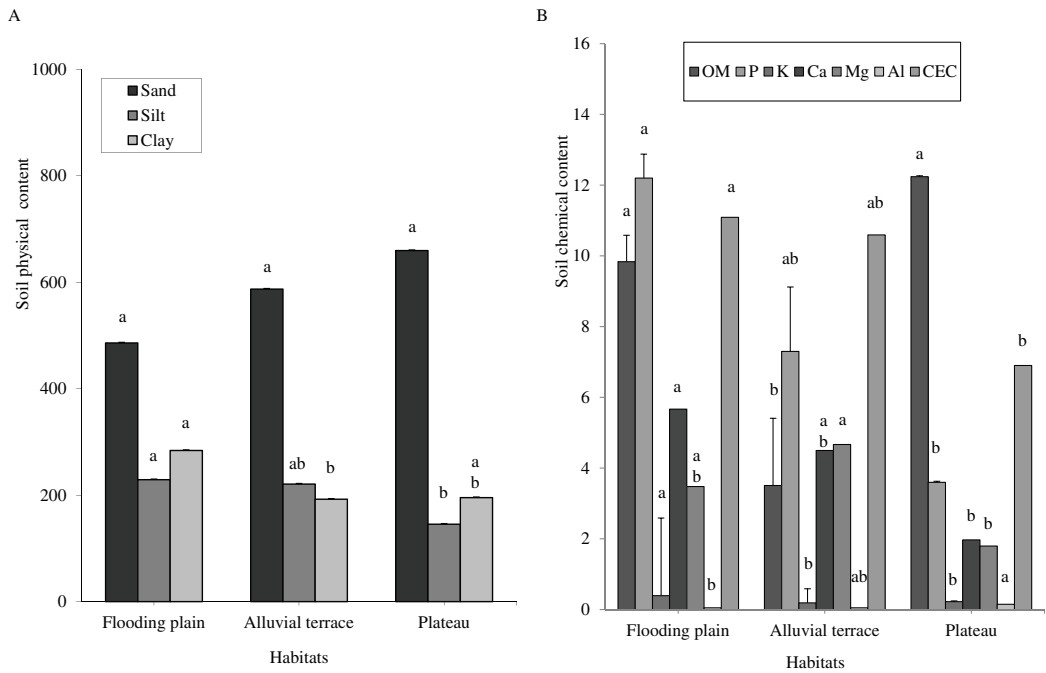

**Figure 2 Tukey's Honest Significant (sand and silt) and Kruskal–Wallis (clay, OM, P, K, Ca, Mg, Al and CEC) tests.** Columns followed by same letter are similar ($p = 0.05$). Error bars indicate standard deviation. Particle size (A) and organic matter (OM) ($g.kg^{-1}$), phosphorus (P) ($mg.dm^{-3}$), potassium (K) ($cmolc.dm^{-3}$), calcium (Ca) ($cmolc.dm^{-3}$), magnesium (Mg) ($cmolc.dm^{-3}$), aluminum (Al) ($cmolc.dm^{-3}$) and cation exchange capacity (CEC) ($cmolc.dm^{-3}$) (B), at 0–20 cm depth in a Caatinga dry forest, Brazil.

**Table 1 Germination (%) (mean ± SD) of *Prosopis juliflora* seeds across three habitats and four seed dispersal treatments in the Caatinga dry forest, Brazil (a total of 12,000 seeds, 10 replicates).**

| Seed dispersal modes | Habitats | | |
| --- | --- | --- | --- |
| | Flooding plain | Terrace | Plateau |
| Seeds on soil surface | 1.3 ± 2.7 c[1]A[2] | 5.3 ± 6.7 cA | 0.7 ± 1.6 bA |
| Buried seeds | 47.4 ± 25.3 aA | 33.4 ± 25.8 aA | 1.6 ± 4.3 bB |
| Cattle-dispersed seeds | 20.5 ± 5.6 bA | 17.2 ± 4.7 aA | 12.9 ± 3.1 aA |
| Mule-dispersed seeds | 13.4 ± 6.1 bcA | 11.7 ± 6.3 bA | 9.2 ± 3.7 aA |

**Notes:**
[1] Means across seed dispersal modes sharing lower-case letters did not differ (Tukey test, $p < 0.001$).
[2] Means across habitats sharing upper-case letters did not differ (Tukey test, $p < 0.001$).

caused by seed burial or dispersal via cattle and mule. Overall, seed dispersal mode was more impacting than habitat for seed germination.

## Seedling survival and life expectancy

In the flooding plain and terrace (Figs. 3A–3D and 4A–4D), *P. juliflora* seedling survival declined steeply in the first 30 days after germination, particularly among seedlings from buried seeds. After this period, survival declined moderately from 30 to 60 days and tended to stabilize 70 days after germination for all seed dispersal treatments. In the plateau, seedling survival tended to stabilize 15 days after germination (soil surface and

**Table 2 Statistical scores of a two-way ANOVA for seed germination in a Caatinga dry forest, Brazil.**
The number of germinated seeds was square-rooted transformed prior analysis.

| Source of variation | F | df | SQ | QM | p |
|---|---|---|---|---|---|
| Seed dispersal mode | 25.5 | 3 | 164.71 | 54.9 | <0.01 |
| Habitat | 38.7 | 2 | 74.87 | 37.43 | <0.01 |
| Interaction | 12.1 | 6 | 102.72 | 17.12 | <0.01 |
| Error | – | 108 | 152.95 | 1.4 | – |

buried seeds), but only at 30 days in the case of seedlings from cattle/mule dispersed seeds (Figs. 5A–5D). Elapsed 180 days after seed germination started, almost all seedlings were already dead in terrace and plateau as seedlings died faster in these habitats, particularly seedlings that emerged from seeds left in the soil surface. In fact, only one seedling from surface seeds (3,000 seeds) achieved 1 year; that is, a seedling in the flooding plain.

Moving to life expectancy, two peaks were observed across seed dispersal modes in the flooding plain. A first peak occurred approximately 75 days after germination, with the exception of seedlings from buried seeds. For these seedlings the first peak occurred at 180 days. The second peak in life expectancy was observed at 195, 210, 240 and 255 days among seedlings from seeds dispersed by cattle, on the soil surface, buried and dispersed by mules, respectively. Life expectancy of *P. juliflora* reached 345 days across all treatments (Figs. 3A–3D; Table S1). In the terrace, a peak of life expectancy occurred within the first 60 days after germination, without further increments from this period onwards. Exception was among seedlings from seeds deposited on the soil surface, whose first peak of life expectancy occurred at 90 day (Figs. 4A–4D; Table S2). The life expectancy in the terrace reached 150 days for all seed treatments, except for those buried in the soil surface, which reached 165 days.

In the plateau, life expectancy among seedlings from soil surface and buried seeds were higher in the first 15 days after germination and tended to decrease consistently from germination to 45 and 120 days of age, respectively. On the other hand, there were peaks in life expectancy at 30 and 60 days among seedlings dispersed via cattle and mules, respectively. However, life expectancy reached 120 and 165 days among seeds buried and cattle-dispersed, respectively (Figs. 5A–5D; Table S3).

## DISCUSSION

The main habitats covered by the Caatinga dry forest differ in soil attributes or conditions. Overall soils along flooding plains and terraces have finer texture and are more humid and fertile than those on the plateau, which are sandy and xeric. In these habitats, both seed germination and seedling survival of *P. juliflora* are relatively reduced. However, the way seeds are dispersed and deposited (i.e., seed dispersal treatments) appears to respond to soil conditions and greatly influences seed germination and early seedling performance across the main Caatinga habitats, although in most of the situations seed germination can be considered low (≤20%) and highly variable. The chances of seedling survival when
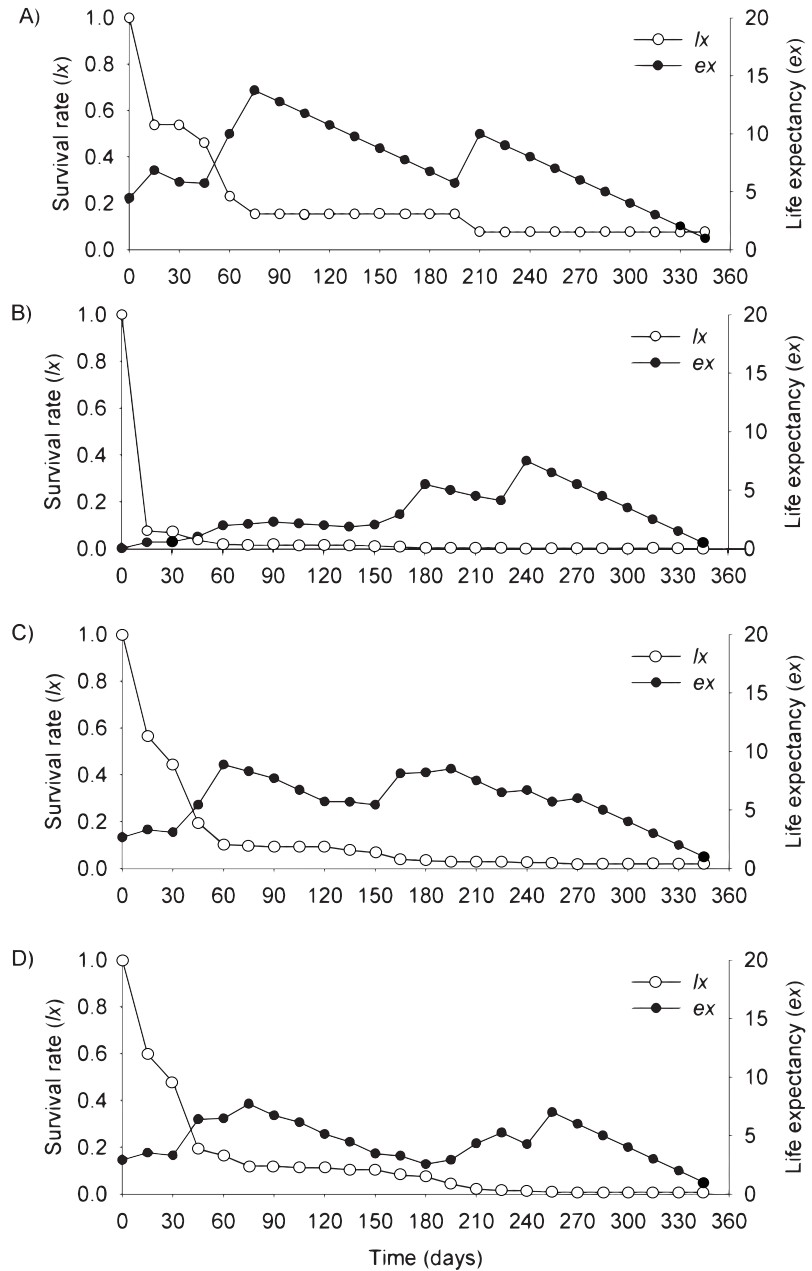

**Figure 3 Survival (*lx*) and life expectancy (*ex*) of *Prosopis juliflora* sown on the soil surface (A), buried (B) or mixed with cattle (C) or mule (D) manure in the flooding plain habitat up to 345 days in a Caatinga dry forest, Brazil.**

coated in manure and across flooding plains are higher, while those deposited on the soil surface on the plateau have a lower survival probability. Thereby seedling life span is longer from those originated with seeds immersed into cattle manure. Moreover, the benefit for seedling dispersal depends on animal species (cattle vs. mule) and the habitat but seedling survival can be extremely low. In synthesis, early seed-seedling fate of *P. juliflora* in the Caatinga dry forest depends on the seed dispersal mode and the habitat seeds are delivered. In other words, interactions between soil attributes and seed dispersal by

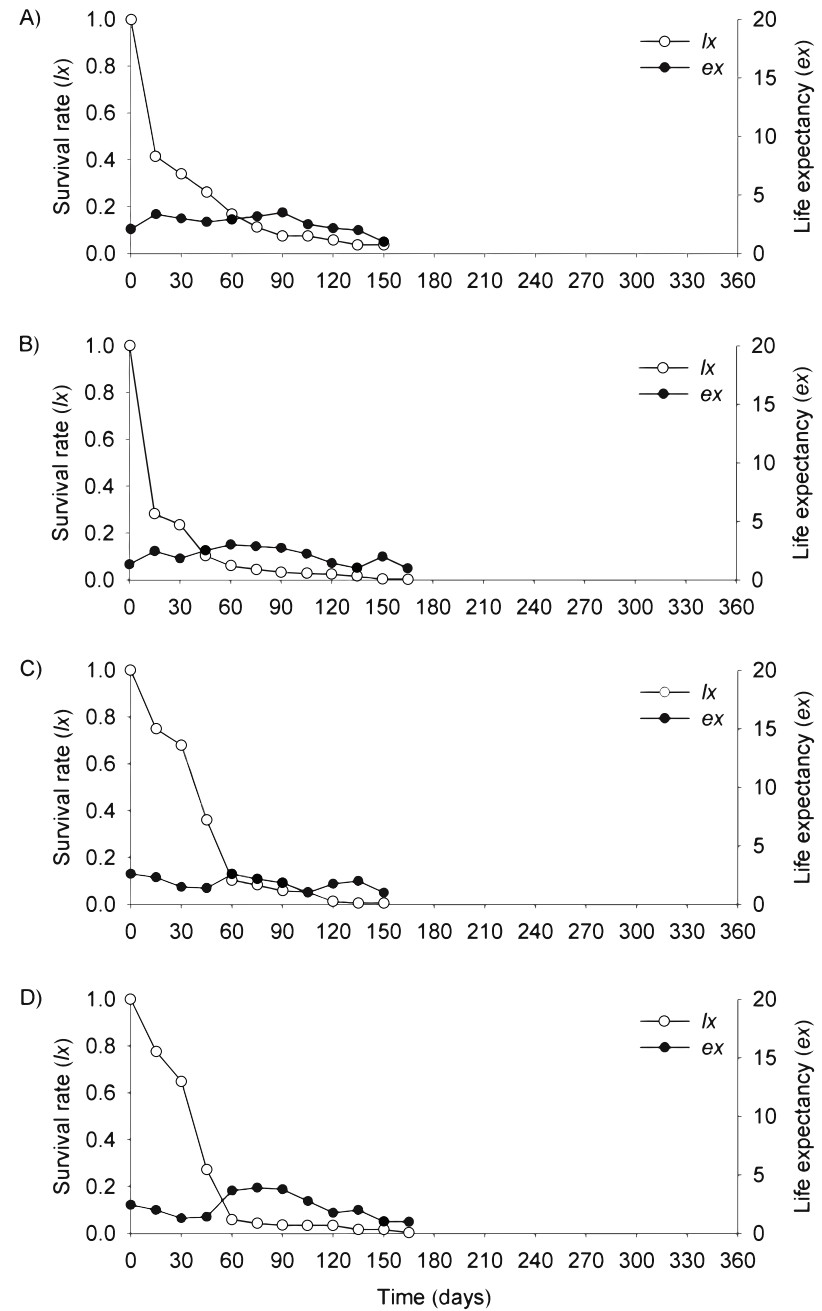

**Figure 4 Survival (*lx*) and life expectancy (*ex*) of *Prosopis juliflora* sown on the soil surface (A), buried (B) or mixed with cattle (C) or mule (D) manure in the alluvial terrace habitat up to 165 days in a Caatinga dry forest, Brazil.**

exotic species apparently define the *P. juliflora* potential as an invasive species across human-modified Caatinga landscapes.

Such a role played by physical conditions has been documented elsewhere. The dominant species *Stipa breviflora* Griseb. (Poales: Poaceae) expansion potential depends on climate changes in Chinese temperate grasslands (*Lv & Zhou, 2018*). Extreme climatic change in Mediterranean rivers is suggested to promote the disappearance of

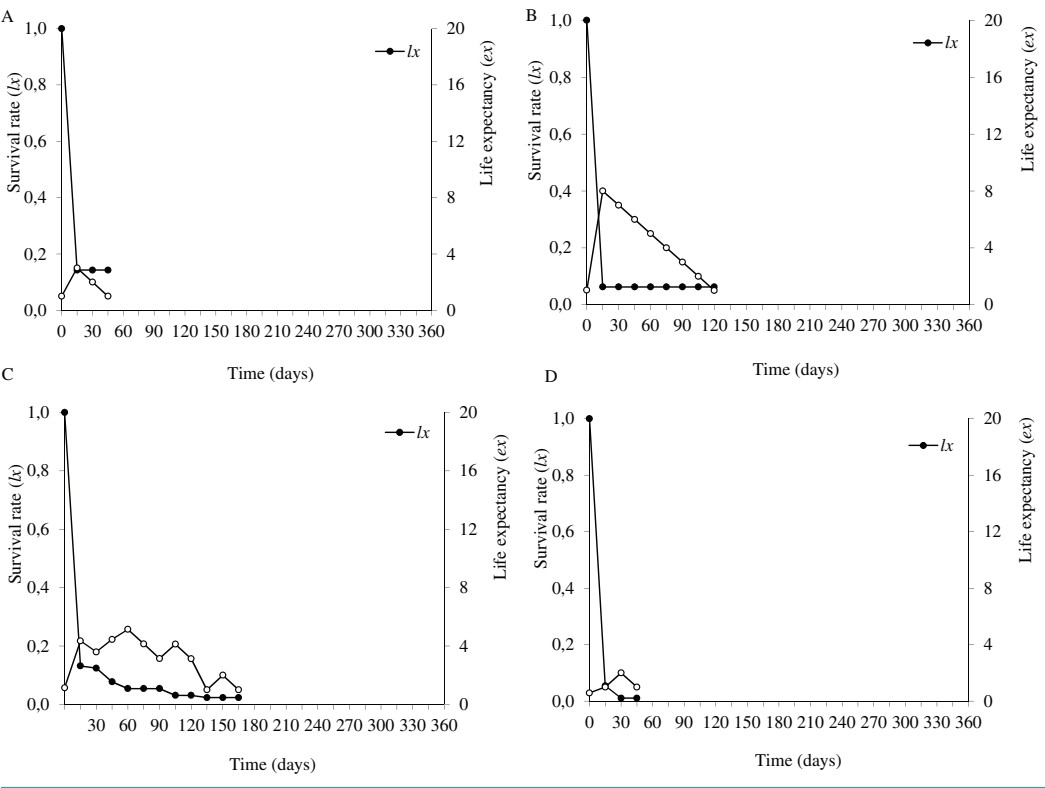

**Figure 5 Survival (*lx*) and life expectancy (*ex*) of *Prosopis juliflora* sown on the soil surface (A), buried (B) or mixed with cattle (C) or mule (D) manure in the plateau habitat up to 165 days in a Caatinga dry forest, Brazil.**

the pioneer and young succession stages of riparian woodlands (*Rivaes et al., 2013*), particularly evident in Europe (*Rivaes et al., 2014*). Moreover, soil differs across habitats, through which *P. juliflora* has been documented to exhibit extreme variations in establishment and stand density in the Caatinga dry forest (*Bailis & McCarthy, 2011*). *Prosopis juliflora* stands have been reported with densities as high as 140.39 stems per ha across flooding plains and alluvial deposits at the Keoladeo National Park in Rajasthan, India (*Mukherjee, Velankar & Kumara, 2017*) and in Tamil Nadu, India (*Gandhi & Pandian, 2014*). Such dense monospecific stands suggest soil conditions and seedling ecology as potential key drivers for this invasive species as previously documented for other regions in which *Prosopsis* species are invasive (*Ansley, Zhang & Cooper, 2018*). *Prosopis* species largely benefit from seed dispersal by domestic animals via dung pellets, improving seed germination and seedling performance and leading to higher life expectancy (*Miranda et al., 2011*, *2014*; *Razanamandranto et al., 2004*). Seeds deposited on the soil surface without manure had the worse seedling performance and survival in our focal landscape; similar to that documented across a variety of habitats and biotas (*Babawi, Campbell & Mayer, 2016*; *Mukherjee, Velankar & Kumara, 2017*).

*Prosopis juliflora* as an invasive plant species benefiting from dispersal services offered by exotic animals is far from novel, with a number of examples for species of this genus (*Abbas, Mancilla-Leytón & Castillo, 2018*). Ingestion and deposition by livestock is not

required for *Prosopis* seeds to germinate, but it improves seed germination, seedling performance and long-distance seed dispersal of this plant as it has a patchy distribution and no native vertebrate species have been found to disperse *P. juliflora* seeds in the Caatinga region. Seed dispersal by native ungulates (i.e., dears, *Mazama americana* Erxleben, 1777 and *Mazama gouazoubira* Fischer, 1814 (Artiodactyla: Cervidae)) is theoretically possible but large vertebrates have been intensively extirpated in the Caatinga region (*Bragagnolo et al., 2019*; *De Oliveira et al., 2019*). Caatinga supports nearly 30 million cattle, goats and mules (*Sampaio & Costa, 2011*; *Santos et al., 2017*) as a large number of active seed dispersers which spread through diverse habitat types, from old-growth forest stands to degraded areas (*Sampaio & Costa, 2011*). Livestock frequently access humid habitats such as river banks on their daily search for fresh water and native forage (*Dias, Massara & Bocchiglieri, 2019*; *Lopes, Montenegro & Lima, 2019*). Moreover, fodder availability is seasonal and scarce (*Costa et al., 2016*), while *P. juliflora* fruits and foliage represent a permanent and appreciable food source for livestock, likely stimulating its fruit consumption.

*Prosopis* has relatively hard pods and its sweet-arillate seeds probably evolved as a response to consumption by native ungulates (e.g., cattle, goats, horses, mules and sheep *Ovis aries* L., 1758 (Artiodactyla: Bovidae)), as an adaptation for contemporary consumption and seed dispersal by domestic livestock (*Nascimento et al., 2014*; *Almeida et al., 2015*). Although a substantial seed portion may be damaged or digested when consumed by livestock, the benefits from dispersal outweigh this drawback. This may be particularly true in the Caatinga as the native flora (mostly abiotically-dispersed) does not benefit from the dispersal services offered by livestock.

Higher nutrient and water content in both flooding plain soils and livestock manure may enhance seedling performance of *Prosopis* species. This was suggested for *Prosopis africana* (Guill. & Perr.) Taub. seeds in sheep and zebu manure in the Sudanian savanna (*Razanamandranto et al., 2004*) and *P. juliflora* in animal manure in the Caatinga in northeast Brazil (*Miranda et al., 2011*, *2014*). Causal mechanisms responsible for differential seedling performance of that plant across seed dispersal modes and habitats were not explicitly examined in our focal landscape. Seeds deposited on the soil surface rarely germinate due to dehydration as observed for *Prosopis ferox* Griseback and *Prosopis flexuosa* DC. across semiarid regions of Argentina and Sudan (*Campos et al., 2011*; *Morandini, Giamminola & Viana, 2013*; *Yoda et al., 2015*). High *Prosopis caldenia* Burkart seed mortality has also been related to soil water availability reduction across semiarid rangelands of Central Argentina (*De Villalobos, Peláez & Elia, 2005*; *De Villalobos & Peláez, 2015*; *Risio et al., 2016*).

Fine-textured soils with greater silt and clay quantities, as observed in the flooding plain, are expected to increase water retention and storage capacity, and thus may increase both seed germination and plant survival. A similar phenomenon was observed for *P. caldenia* and *P. glandulosa* since allometric growth and survival was directly proportional to the water availability along a topographic gradient in the Chihuahuan Desert, Mexico, in the semiarid rangelands of Central Argentina and in the arid and semiarid ecosystems of the USA, respectively (*Martínez & López-Portillo, 2003*; *De Villalobos, Peláez & Elia, 2005*;

*Maestre & Reynolds, 2006*). Water storage capacity (*Arcone et al., 2008*) and organic matter, phosphorus and calcium accumulation are usually higher in fine-textured soils with greater silt and clay quantities, found mainly in gallery forests (*Jiménez et al., 2008*), which gives these soils a greater CEC (*Shiferaw et al., 2004*; *Silva et al., 2008*). In synthesis, *P. juliflora* seeds and seedlings are sensitive to water stress and habitat desiccation.

The higher nutrient availability in the flooding plain and terrace, along with the low aluminum content in the flooding plain, may also be important because excess of this element in the soil can impair the calcium, magnesium, phosphorous and potassium absorption by *Prosopis* plants, as reported for *P. juliflora* in the Caatinga (*Jiménez et al., 2008*; *Oliveira et al., 2017*). However, calcium and magnesium counteract this detrimental effect as observed for *Prosopis alba* Griseb., *Prosopis pubescens* Benth. and *Prosopis ruscifolia* G. in the central and western Gran Chaco phytogeographical region, Argentina, and in El Paso, TX, USA, respectively (*Velarde, Felker & Gardiner, 2005*; *Meloni, 2012*; *Zapala et al., 2014*). The soil nutrient availability improved plant vigor, making them less vulnerable to drought stress in our focal landscape is similar to that reported for *Prosopis cineraria* (L.) Druce (Ghaff) in United Arab Emirates (*Gil & Al-Shankiti, 2015*; *Song, Li & Hui, 2017*). Nutrient-rich soils increase the *P. juliflora* seedling establishment, allowing its embryos to grow quickly (*Nasr et al., 2012*; *Patil & Karadge, 2012*; *El-Keblawy & Abdelfatah, 2014*).

The high nutrient and water availability in animal manure also apparently create suitable microhabitats for seeds and seedlings. This substrate is favorable during the critical germination and emergence periods, especially considering the spatially erratic rainfall in the Caatinga even in the rainy season (*Sampaio & Costa, 2011*). Such services provided by livestock manure benefit the *Prosopis* seeds of different species from South America to the Middle East and Africa (*Majd et al., 2013*; *Westphal et al., 2015*; *Araujo, Pérez & Bonvissuto, 2017*). On the other hand, the rapid dehydration and hardening of animal manure reduced seed germination of *P. glandulosa* in the arid and semiarid regions of northeastern Mexico and in the drylands of Japan (*Garza et al., 2013*; *Abdalla et al., 2017*). The lower soil moisture in the plateau of the Caatinga accelerated the drying and hardening process of cattle and mule manure (C.E. de Souza Nascimento, 2018, personal observation). This adversely affected the *P. juliflora* germination and emergence and contributed to its lower survival and life expectancy compared to the flooding plain and terrace habitats.

The higher survival rate and life expectancy of *P. juliflora* seedlings germinated from seeds mixed with cattle manure compared to those mixed with mule manure in the plateau can be attributed to differences in the composition of these animal dungs (*Campos et al., 2008*; *Nascimento et al., 2014*). Mule manure has a lower water content and higher fibrous sheath content encasing seeds (*Gonçalves et al., 2013*). The fibrous sheath covering the seeds probably reduces seed imbibition due to the fact that these animals are not ruminants like cattle (*Gonçalves et al., 2013*). Both exotic and native ungulates can favor invasive plant species by (1) creating disturbed habitats via feeding, trampling and movement, (2) controlling or eliminating palatable native species and (3) dispersing seeds via endozoochory and epizoochory (*Vavra, Parks & Wisdom, 2007*). The colonization and

direct seed dispersal services provided by domestic ungulate livestock was confirmed, and evidences for improved germination and seedling performance associated with fruit consumption and the seed dispersal, coated in protective manure, to suitable habitats was also confirmed. However, even in this favorable context few seedlings appear to survive longer and move to other life stages, suggesting that *P. juliflora* invasion capacity in the Caatinga dry forest relies on intense seed production and dispersal on suitable habitats.

Caatinga dry forest has been converted to small-scale farming for the agriculture and livestock subsistence, as found in other dry forests and savannas globally (*Miccolis et al., 2017*; *Pérez-Marin et al., 2017*). Farming has a continuous demand for the intentional introduction of useful alien species such as cattle, goats and the multiple-use shrub *P. juliflora*, which has now established throughout the Caatinga (*Nascimento et al., 2014*; *Almeida et al., 2015*; *Ilukor et al., 2016*; *Walkie et al., 2016*). Seed and seedling sensitivity to water stress and desiccation limits the *P. juliflora* proliferation and restricts its establishment to humid/more fertile soils even when its seeds are coated in livestock manure. *Prosopis juliflora* has spread throughout the Caatinga region establishing monospecific stands across flooding plains, river banks and sedimentary deposits despite such a life-history limitation. This invasion relies on a close mutualism between human populations and their commensals as agriculture provided degraded habitats, affect soil conditions across river banks via deposition of fine material, while livestock provides seed dispersal services due to the presence of arillated and hard-coated seeds; that is, a high-quality seed dispersal service that is not available for the native flora in the Caatinga, thereby conferring an adaptive advantage to *Prosopis* species. *Prosopis juliflora* is expected to spread in the Caatinga with dense populations locally because the rural populations continue to convert Caatinga dry forest into degraded habitats (*Sfair et al., 2018*). The mutualism with human populations favors *P. juliflora*, as found for other invasive *Prosopis* species across semiarid rangelands globally (*Busso, Bentivegna & Fernández, 2013*). A general perspective on how some *P. juliflora* life-history traits confer invasive capacity has been described (at least relative to early establishment), but other traits (e.g., resprouting capacity, seedling tolerance to herbivory and trampling, allelopathic abilities) enabling *P. juliflora* to become a human commensal deserves further investigation.

## CONCLUSIONS

The *P. juliflora* invasion capacity promoted by patterns of seed germination and seedling survival depends on habitat and seed dispersal mode in the Caatinga dry forest as this species benefits from human-disturbed, humid soils and manure-involved seeds provided by livestock. This perspective highlights plant-animal interactions among exotic species as potential driver for successful invasion, proliferation and habitat distribution of invasive plants in human-modified landscapes or rangelands.

## ACKNOWLEDGEMENTS

To Asia Science Editing of Republic of Ireland and Kieran Withey (Lancaster University in Lancaster, United Kingdom) for English editing and correction of an early and final

version of this manuscript, respectively. Three anonymous Referees offered constructive criticism.

### Funding

Marcelo Tabarelli's research in the Caatinga dry forest was supported by the Alexander von Humboldt Foundation (Germany). These Brazilian institutions provided financial support: "Conselho Nacional de Desenvolvimento Científico e Tecnológico (CNPq)", "Coordenação de Aperfeiçoamento de Pessoal de Nível Superior (CAPES)", "Fundação de Amparo à Pesquisa do Estado de Minas Gerais (FAPEMIG)" and "Programa Cooperativo sobre Proteção Florestal (PROTEF)" of the "Instituto de Pesquisas e Estudos Florestais (IPEF)". The funders had no role in study design, data collection and analysis, decision to publish, or preparation of the manuscript.

### Grant Disclosures

The following grant information was disclosed by the authors:
Alexander von Humboldt Foundation (Germany).
Conselho Nacional de Desenvolvimento Científico e Tecnológico (CNPq).
Coordenação de Aperfeiçoamento de Pessoal de Nível Superior (CAPES).
Fundação de Amparo à Pesquisa do Estado de Minas Gerais (FAPEMIG).
Programa Cooperativo sobre Proteção Florestal (PROTEF).
Instituto de Pesquisas e Estudos Florestais (IPEF).

### Competing Interests

Wagner de Souza Tavares is employed by Asia Pacific Resources International Holdings Ltd. (APRIL), PT. Riau Andalan Pulp and Paper (RAPP).

### Author Contributions

- Clóvis Eduardo de Souza Nascimento conceived and designed the experiments, performed the experiments, analyzed the data, prepared figures and/or tables, authored or reviewed drafts of the paper, and approved the final draft.
- Carlos Alberto Domingues da Silva conceived and designed the experiments, performed the experiments, analyzed the data, prepared figures and/or tables, authored or reviewed drafts of the paper, and approved the final draft.
- Inara Roberta Leal conceived and designed the experiments, performed the experiments, analyzed the data, prepared figures and/or tables, authored or reviewed drafts of the paper, and approved the final draft.
- Wagner de Souza Tavares conceived and designed the experiments, performed the experiments, analyzed the data, prepared figures and/or tables, authored or reviewed drafts of the paper, and approved the final draft.
- José Eduardo Serrão conceived and designed the experiments, performed the experiments, analyzed the data, prepared figures and/or tables, authored or reviewed drafts of the paper, and approved the final draft.

- José Cola Zanuncio conceived and designed the experiments, performed the experiments, analyzed the data, prepared figures and/or tables, authored or reviewed drafts of the paper, and approved the final draft.
- Marcelo Tabarelli conceived and designed the experiments, performed the experiments, analyzed the data, prepared figures and/or tables, authored or reviewed drafts of the paper, and approved the final draft.

## Data Availability

Raw data is available in the Supplemental Files.

## Supplemental Information

Supplemental information for this article can be found online at http://dx.doi.org/10.7717/peerj.9607#supplemental-information.

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
