# Peer review of "Seed germination and early seedling survival of the invasive species Prosopis juliflora (Fabaceae) depend on habitat and seed dispersal mode in the Caatinga dry forest"

_PeerJ, doi:10.7717/peerj.9607_

## Round 0.1 · original submission · Major Revisions

I have now received comments from 3 expert reviewers. All 3 of the reviewers are in agreement that this is an interesting and well-designed study that will be of interest to readers. Key among the reviewers' comments, is the need for additional English language and grammar edits. All 3 have offered assistance in this regard, with Reviewer 1 providing many useful edits in a PDF copy. The reviewers also suggest a more concrete explanation of the research goals and basis for the work in the introduction and some additional discussion topics.

While the comments are extensive, they reviewers have offered clear advice and I recommend responding to each point when possible. Finally, reviewer 3 offers a suggestion for a mixed modeling approach in their comments. If this is possible, I agree with reviewer 3 that being able to better parse the importance of soil properties would add to your manuscript. The manuscript will benefit from following the reviewers' advice and I look forward to your resubmission.

Reviewer 1 ·

Basic reporting

The main conclusion is clear and supported by the results. The structure and general language of the article are also clear. I note that the authors have used an English editing and correction service, but I suggest that this service has not done its job: there are many small grammatical errors throughout the manuscript. As a native English speaker, I have taken the liberty of correcting these on the pdf (please check the attached pdf file), so this should not be a problem for a final published version. The literature is well referenced (although I have suggested one possible paper that could be relevant in the Introduction – see below), and figures are appropriate. The Supplementary materials include data for survival, mortality, life expectancy and age structure.

Experimental design

The study represents original primary (experimental) research and does fall within the aims and scope of the journal. The research question is timely and the study is a useful investigation based on a straightforward experimental design, but applied in specific circumstances. The data collection, statistics and presentation are all of a high standard and the Methods section includes details that would allow the study to be replicated.

Validity of the findings

The findings are valid and supported by the research.

Additional comments

The Introduction includes background information on the fact that plant life history traits are key to the ‘invasiveness’ of invasive species, and here there is a strong emphasis on studies that show the importance of reproductive traits. It might be useful to cite a recent study (Guo et al. (2018) Ecology Letters 21(9), 1380–1389) showing that, world-wide, the naturalization or long-term establishment of invasive species depends on the overall suite of traits that comprises the general functional strategy, and thus including vegetative traits and the integration of adaptive traits in general.

In the Results section (lines 244-245) it might be useful to add that the soil organic matter content at the plateau was extremely variable.

The full latin binomials for cattle and mule are presented in the Results section (lines 249 to 251), but they should be presented at the first mention of these organisms in the Materials and Methods section.

Annotated reviews are not available for download in order to protect the identity of reviewers who chose to remain anonymous.

Reviewer 2 ·

Basic reporting

The authors have presented a good background on this topic. The language use is professional, and the paper is structured conform journal guideline.

Experimental design

The manuscript is within the journal’s scope. It has well-defined research questions considering the current knowledge gap in the literature, followed by a rigorous investigation. Methods are well described however I would recommend explaining the method through the flowchart but in a more symbolic way, not just text.

Validity of the findings

Data and statistical analysis are robust and controlled.

Additional comments

Authors of this manuscript have presented a study concerning the invasive Prosopis juliflora (Fabaceae) seedling survival and growth depend on habitat and seed dispersal mode in the Caatinga dry forest. The authors have found that the plant-animal mutualism between exotic species represents a driving force to spreading P.juliflora across riparian forests, degraded river banks and sedimentary deposits, while this plant fails to dominate dryer habitats in the Caatinga dry forest. The topic is interesting, an important issue and generally well written, well structured and contributes to the existing knowledge. However, there are still some occasional grammar errors through the manuscript especially the article ‘’the’’, ‘’a’’ and ‘’an’’ is missing in many places, please make a spellchecking.

The results and discussion section needs further improvement, compare your findings with the other author's conclusions.
• In general, the manuscript needs to revised again concerning technical and grammae errors.
• Please provide more deep discussion about your results, compare your findings with the other author findings.
• Please clearly state the novelty of this work.
• Please check the reference style, some of the references are not according to the journal style, especially the journals abbreviations.
• At present, neither the original contributions of the paper nor its practical significance is clear for me.
• As said, it could be a failure to comprehend the main argument; and lightening the discussion would enhance comprehensibility and make the authors’ point clearer and stronger.
• Therefore, the reviewer recommends to further improve the manuscript before accepting it for publication. Some of the specific comments are listed below.
• The conclusion section is to short, please elaborate it a little bit more.
• Please consider citing following literature; reviewer believes that it may help the authors to have a more holistic overview about climate impact on plants:

Rivaes, R., Rodríguez‐González, P. M., Albuquerque, A., Pinheiro, A. N., Egger, G., & Ferreira, M. T. (2013). Riparian vegetation responses to altered flow regimes driven by climate change in Mediterranean rivers. Ecohydrology, 6(3), 413-424.
Lv, X., & Zhou, G. (2018). Climatic Suitability of the Geographic Distribution of Stipa breviflora in Chinese Temperate Grassland under Climate Change. Sustainability, 10(10), 3767.
Rivaes, R. P., Rodríguez-González, P. M., Ferreira, M. T., Pinheiro, A. N., Politti, E., Egger, G.,& Francés, F. (2014). Modeling the evolution of riparian woodlands facing climate change in three European rivers with contrasting flow regimes. PloS one, 9(10), e110200.

Concluding Remarks
The work presented in this manuscript is an interesting topic, it needs some more efforts to improve it further. Reviewer recommend major revision of this manuscript and publishing it only after specific improvement of the current version are made.

Reviewer 3 ·

Basic reporting

• Clear and unambiguous, professional English used throughout

The authors set out to conduct an experiment looking at the effects of livestock and local geomorphology/soil chemistry on seed germination, seedling survival and a slightly ambiguous estimate of “life expectancy” of Prosopis juliflora in three different alluvial habitat’s in the São Francisco River valley of the semi-arid Caatinga region of Brazil. This paper can be greatly improved by consulting/editing by a native/fluent English writer. As it stands now, the introduction and stated purpose of the study are difficult to comprehend.

• Literature references, sufficient field background/context provided

The article does include an extensive review and background of the topic. However, most of this is contained in the discussion. It would be good to include more background information in the introduction. It would be good for the introduction to include a more in-depth overview of the Caatinga region in general. What are the native plant communities that have been replaced by P. juliflora invasions? Give specific examples, including dominant genera that have been excluded. It would also be good to include some general information on the types and history of the human disturbance in the region. The authors do mention that P. juliflora was introduced as a food source for livestock to the region in the 1940s. This is good, but that is far as they go in describing the extent and degree of the disturbance. If there has been sustained cattle grazing since the 1940s, what percent of the Sao Francisco river valley native vegetation has been replaced by invasive P. juliflora? Lines 132:133 (word doc) states “The Caatinga vegetation has increasing human disturbance levels associated to agriculture and livestock production by low-income rural populations (da Silva et al., 2014; Leal et al., 2014; Ribeiro et al., 2015).” This suggest that other agricultural practices besides grazing are active in the area. What are they? Do these co-occur spatially with grazing? Or is grazing limited to rangelands? Are there other land management practices that occur? Is fire used? A more complete description of disturbance and the role of grazing should be reviewed.

• Professional article structure, figs, tables. Raw data shared.

Supplementary_material_(3,4, 5) contained counts of living P. juliflora for each sampling period at each of the habitat types. The authors need to check that column headings are consistent across each table. Supplementary_material 3 differs from 4 and 5. No raw data was included for the soil data. Only one profile example was included for one of the sites. However, the methodology has no mentions of topographic surveys. This data appears to have come from another source? There is no raw data reported for soil water content either. The methodology report that soil moisture content was measured every 2 weeks over an 11 month period. This time series data is not shared.

The figures and tables in the main manuscript body are professional and match what is described in the main text.

• Self-contained with relevant results to hypotheses.

The study appears to be self-contained and not inappropriately subdivided to increase publication count.

Experimental design

• Original primary research within Aims and Scope of the journal

This question that this manuscript sets out to address is within the Aims and Scope of the journal

• Research question well defined, relevant & meaningful. It is stated how research fills an identified knowledge gap.

The research goals need to be more explicitly stated. This perhaps something that can be clarified with the assistance of more fluent writing. I am uncertain about the emphasis on “life expectancy”. The authors need to do a better job of defining what they mean by life expectancy in the context of their study. How is this any more relevant than seedling survival? Life expectancy is typically measured as the average number of years an individual will live. Once established, most Prosopis are relatively long-lived species. For example Bhojvaid et al. 1996. Reclaiming sodic soils for wheat production by Prosopis juliflora.. report on 30 year old P. juliflora plantations. They probably live much longer. Life expectancy seems misplaced for one year of monitoring of seedlings.

• Rigorous investigation performed to a high technical & ethical standard

The investigation was conducted to a high technical standard.

• Methods described with sufficient detail & information to replicate

Sampling methods could be explained with more clarity. Line 188:191. “The moisture content of 10 soil samples (determined using the gravimetric method) was measured twice a week at 20 cm depth from March 2014 to February 2015 and it was analyzed according to soil type and environment (Francesca et al., 2010).” Is this 10 soil samples overall, or 10 soil samples per habitat type? It is not clear.

Lines 193:194 No need to have the same calculation repeated twice. Simply list which version of H was used.
Line 199:200 What are Brazilian standard protocols? Please briefly state if these are different than others.
Line 201:202. How were seeds collected and when? Are they mature fruits taken off of plants or taken from the ground? Livestock treated seeds were only searched for in manure? Or were they purposefully fed to livestock first and then collected in manure? Please be more explicit.
Please clarify the sowing protocol. Lines 215:216 “One hundred seeds were adopted per plot, totaling 1,000 seeds per treatment, in mid March 2014 (i.e. raining season).” Did each plot contain all 4 treatments? 100 seeds per plot across 120 plots would be 12,000 seeds. You have 4 treatments. That's 3,000 seeds/treatment and 1000 seeds per treatment for each habitat ( Floodplain, Alluvial, Plateau).
There should be 25 seeds for each treatment per plot. Please state numbers more clearly

Lines 220:221 “The seeds from the fruits and in the dung were collected from several trees and animals, respectively, in the year that P. juliflora trees fruited/flowered.” This seems to imply that P juliflora doesn’t fruit or flower every year? If that is the case it is directly relevant to the study of seed dispersal and recruitment. Or does this just mean that seeds were collected in the year preceding the study? Please clarify.

Lines 238. Equation for Tx showup on ms word document but not on pdf. Please be aware of this.

Lines 230:240 Life expectancy and survival tables. Be mindful of your variable descriptions. Each one contains an x which is confusing because it could mean multiply. I’m not sure it’s intended to be used this way Example Line 235: number of individuals dead during the age class x (dx)

Validity of the findings

The authors have a rich dataset that contains local soil chemistry measurements as well as time series on soil moisture, germination and seedling survivorship. The statistical analysis were fine. However, I wonder why the authors focus on life expectancy.

Why not report or present in figures germination (#s and %) and seedling survival (#s and %)? These are data that are buried in the supplemental material but seem to be the most relevant data to the stated research questions.
Also, survivorship across all treatments was low.


Also, soil moisture was collected as a time series. In semi-arid environments water is often a limiting factor for survivor ship. The timing of dry down could be directly related to timing of mortality in each of the treatments. It would be good to visually present the soil moisture time series data in contrast to time series of seedling survivorship.

Overall, seedling survival was extremely low across all treatments. According to the supplemental tables the highest number of living individuals at the end of the study was 4. These were in the floodplain x cattle treatment. 4 out 1000 seeds seems like a particularly relevant point that should be made. There are other influence besides dispersal alone that promote the expansion or maintenance of native plant communities. For e.g., see Weltzin et al. 1997 SMALL‐MAMMAL REGULATION OF VEGETATION STRUCTURE IN A TEMPERATE SAVANNA. Small mammals and other herbivores can maintain existing plant communities preventing colonization of Prosopis. Is there a possibility that similar occurrences are occurring in the Caatinga? Just something to think and perhaps address in the discussion.

Additional comments

Overall, I think the study design is valid. Questions that move beyond the strict physical parameters of species expansions and into the biological components are important additions, and it's nice to see that folks are working on these questions. I truly feel that all the major components of the story the authors are trying to tell are here. I do feel that the presentation of their data can be improved. It would be nice to see seedling survival/germination over time by habitat type and soil moisture over time by habitat type.

Also, can the spatial resolution of the soil characteristics, be tied to individual plots, or blocks of plots. If so, this analyses would be greatly improved by adopting a mixed modelling approach. It appears from the large drop in germination at the plateau sites that soil types are an important piece of the story. With 120 plots you may have enough to start to tease apart the general characteristics that promote Prosopis germination and establishment. This could further parameterize where Prosopis may be more likely to invade into the future.

---

## Round 0.2 · Major Revisions

The reviewers have found your revised manuscript to be a great improvement and have generally favorable opinions regarding the work. Reviewer 3 raises several important consideration that I would like you to address at this time. Please respond to this Reviewer's comments in a line by line response. I look forward to receiving your revision.

Reviewer 1 ·

Basic reporting

The quality of the English has been much improved since the last submission. The extra publication I suggested has been cited.

Experimental design

no comment

Validity of the findings

no comment

Additional comments

I am satisfied that the authors have adopted all of my suggestions for improvement based on the previous submission.

Reviewer 2 ·

Basic reporting

The manuscript is improved significantly.

Experimental design

The manuscript is improved significantly.

Validity of the findings

The manuscript is improved significantly.

Additional comments

The manuscript is improved significantly. Accept!

Reviewer 3 ·

Basic reporting

The language in the manuscript has greatly improved since the previous version. While there are still occasional errors (Please check spelling. The focal species is reported as both Prosopis and Prosopsis multiple times) it reads reasonably well.

The authors do give a reasonable historical background and context to their experiment, and the literature references for this paper are appropriate and sufficient for the manuscript.

Please include units in your figures. For example – Figure 3. Is Field Capacity being reported here as %, g/g? Gravimetric water content should be reported as θ = cm^3 cm^-3. Are the units of life expectancy days?

I have specific concerns on what is being reported in results of the manuscript. The author’s stated objectives are to evaluate seed germination, seedling survival, and life expectancy (lines 142 – 144). However, seed germination is never reported in the results of the paper. They are in the supplemental tables. I gather that germination was the raw number of germinates and is reported as a survival rate at T=0 on the figures, however this is not clear. A table showing the total number and percent that germinated for each experimental category would be a simple and clear way to report germination.

Why are survival rates being reported with values much higher than 1 (100%). Is this because at each time interval it’s a step function of the total available seedlings in the previous time period? If germination continued from 0-30 days this is why your “rate” of survival increases over time. The figures (3-5) could more clearly express the data if % survival with survivorship curves were presented for each experimental group. There are many examples of this. See Salihi and Norton. 1987. Survival of Perennial Grass Seedlings Under Intensive Grazing in Semi-Arid Rangelands. Journal of Applied Ecology, Vol. 24 (1) for a good example.

I believe the authors are correct in stating that water availability is an important factor in determining seedling survival. The reporting could be greatly improved if soil moisture measurements were reported in time-series to match their reporting of survivorship (could include germination too).

Experimental design

This study has a simple and straightforward experimental design; to evaluate seed germination, seedling survival and life expectancy across different geomorphologies and pre-germination treatments. The authors do a reasonable job addressing this objective in their experimental design and their experimental methods are clear enough to replicate.

The authors could be more rigorous and relate the potential covariate data they collected (soil moisture and nutrient data) in a statistical way (GLMs) to seedling survivorship. It appears that the authors setup this study with intent to do so, but is not presented in this manuscript.

Validity of the findings

Seed germination and seedling survival are appropriate measurements to report here. I worry about the estimates of life expectancy. Prosopis can be long lived species. Estimating the life expectancy based on 1 year of seedling survival might not be appropriate.

Seed germination and seedling survival are appropriate measurements to report here. I worry about the estimates of life expectancy. Prosopis can be long lived species. Estimating the life expectancy based on 1 year of seedling survival might not be appropriate.

The supplemental data appears to suggest that maximum number of seedlings still surviving after 1 year was 4, that were embedded in cow manure in the floodplain. The low overall survivorship should be addressed in the discussion.

Additional comments

Objectives of the paper are clear. The authors seek to evaluate if geomorphological surface and the influence of livestock play a disproportionate role in seed germination and seedling survival of invasive Prosopis juliflora in the Caatinga dry forest. Their overall experimental design and the implementation of the experiment seem to do a reasonable job addressing this objective. I believe where this manuscript could be improved is in the reporting of their findings. Starting with variation in seed germination. This is a topic that may be of interest across a large number of people studying Prosopis in general, and is particularly relevant to the study in this manuscript. I also feel that presenting simple survivorship curves for the seedlings instead of the life expectancy/survivor rate figures would be a visually strong way to present the data and tell the story.

---

## Round 0.3 · accepted · Accept

I appreciate your efforts toward addressing the comments provided by the reviewers. I am certain that your article will be of interest to our ecologically focused readers and I hope you receive feedback from them in the future.